# Demethylation of methylguanidine by a stepwise dioxygenase and lyase reaction

Malte Sinn [1] ✉, Dietmar Funck [1], Felix Gamer [1], Clemens Blumenthal[1], Cecilia Kramp[1] & Jörg S. Hartig [1,2] ✉

Guanidine-responsive riboswitches control genes that enable either detoxification or assimilation of guanidino compounds. In *Vreelandella boliviensis* and other halophilic bacteria, genes encoding the guanidine carboxylase pathway are found in a guanidine riboswitch-regulated operon, along with two uncharacterized genes annotated as 2-oxoglutarate (2-OG/Fe(II))-dependent dioxygenase family protein and hypothetical protein, respectively. Here we show that the 2-OG/Fe(II)-dependent dioxygenase efficiently hydroxylates methylguanidine. The resulting N-(hydroxymethyl)guanidine constitutes an unexpectedly stable hemiaminal that slowly decays to guanidine and formaldehyde. The second protein strongly accelerates the fragmentation of N-(hydroxymethyl)guanidine into guanidine and formaldehyde, thus acting as N-(hydroxymethyl)guanidine lyase. Interestingly, the class II guanidine riboswitch in front of the guanidine carboxylase gene does not discriminate between guanidine and methylguanidine, whereas the guanidine class I riboswitch at the start of the entire operon is specific for guanidine. *V. boliviensis* exhibits growth in minimal media with either guanidine or methylguanidine as sole nitrogen source. Comparative proteome analysis revealed that the entire guanidine carboxylase operon is strongly expressed under these conditions. The presented study broadens our understanding of guanidine metabolism by describing two enzymatic activities that jointly catalyze the demethylation of methylguanidine.

The occurrence of guanidine in natural samples had been reported sporadically, but its biological roles remained underappreciated until the discovery of riboswitches that respond to this small, nitrogen-rich molecule. So far, four distinct classes of riboswitches have been described that induce gene expression in response to the presence of guanidine[1–4]. The wide-spread occurrence and evolution of different guanidine riboswitches in a broad range of bacterial taxa demonstrates that it seems beneficial for many bacteria to monitor intracellular guanidine levels. The initial discoveries of guanidine riboswitches were followed by a series of investigations of the functions of riboswitch-regulated genes in guanidine metabolism: In many bacteria, specific transporters are induced that export guanidine from the cytoplasm[5].

Additionally, two separate guanidine degradation pathways for nitrogen assimilation, frequently associated with putative uptake systems for guanidine or related compounds, were discovered. With guanidinium hydrolase, a $Ni^{2+}$-dependent enzyme from the ureohydrolase family was described that directly hydrolyzed guanidine to urea and ammonia[6,7]. A homologous enzyme is used by ammonia-oxidizing bacteria that were capable of using guanidine as the sole source of nitrogen and energy to support growth and carbon assimilation from $CO_2$[8]. In the guanidine carboxylase (Gca) pathway, guanidine is activated by ATP-dependent carboxylation followed by hydrolysis of carboxyguanidine via allophanate to $CO_2$ and ammonia[2,9,10]. Recent publications have expanded the role of guanidine, indicating that it

[1]Department of Chemistry, University of Konstanz, Konstanz, Germany. [2]Konstanz Research School Chemical Biology (KoRS-CB), University of Konstanz, Konstanz, Germany. ✉e-mail: malte.sinn@uni-konstanz.de; joerg.hartig@uni-konstanz.de

can serve as signal for the presence of the plant toxin canavanine, the 5-oxa-derivative of arginine, which protects plants from herbivores. Canavanine hydrolysis by arginase gives rise to the toxic degradation product canaline, which can be prevented by cleavage of canavanine into homoserine and hydroxyguanidine by canavanine-γ-lyase[11]. Canavanine by itself is also toxic via misincorporation into proteins. The Gd-IV riboswitch-associated tRNA editing factor CtdA specifically hydrolyzed mischarged canavanyl-tRNA$^{Arg}$, thus preventing the misincorporation of canavanine into the proteome[12].

Although there is a growing body of knowledge about the functions of guanidine in nature, the number of enzymatic reactions that produce guanidine is limited. Recently, we demonstrated that members of clade 23 of 2-OG/Fe(II)-dependent dioxygenases in plants hydroxylate arginine or homoarginine at the C5- or C6-position, resulting in a spontaneous decay of the hemiaminal to glutamate semialdehyde or aminoadipate semialdehyde, respectively, and free guanidine[12]. The ethylene-forming enzyme, a related 2-OG/Fe(II)-dependent dioxygenase from fungi and plant-pathogenic bacteria that produces ethylene from 2-oxoglutarate, also releases guanidine by C5-hydroxylation of arginine in sub-stoichiometric amounts[13–16]. A third enzyme of the same enzyme family, NapI, catalyzes the 3,4-desaturation of arginine[17]. NapI also hydroxylates arginine at the C5 position in a minor side reaction that yields guanidine. In general, enzymes of the 2-OG/Fe(II)-dependent dioxygenase family catalyze a broad range of hydroxylation or desaturation reactions[18]. In this enzyme family, binding of molecular oxygen to the Fe(II) reaction center is followed by oxidative decarboxylation of 2-OG and typically the formation of a reactive Fe(IV)-oxo intermediate[19]. This activated oxygen atom subsequently abstracts a hydrogen atom from a C-atom of the substrate to induce hydroxylation, halogenation, desaturation, or other reactions. Two sub-classes of the 2-OG/Fe(II)-dependent dioxygenase family, protein methyllysine demethylases and nucleic acid demethylases, hydroxylate methyl groups in their substrates[20]. The resulting hemiaminals are unstable in aqueous solution and spontaneously decompose to formaldehyde and the de-methylated substrates[21,22].

Here, we describe the discovery of a 2-OG/Fe(II)-dependent dioxygenase encoded in the genome of the halophilic bacterium *Vreelandella boliviensis* LC1 (syn. *Halomonas boliviensis* LC1). The genome of *V. boliviensis* contains an operon comprising genes for the guanidine carboxylation pathway as described above (Fig. 1). Like in several other bacteria, this operon is preceded by a guanidine class I riboswitch and additionally contains a guanidine class II riboswitch in front of the *gca* gene[2]. In *V. boliviensis*, the operon is extended by genes encoding a putative 2-OG/Fe(II)-dependent dioxygenase (Uniprot Acc. No. A0A265DXP8) and a hypothetical protein (Uniprot Acc. No. A0A265DXW6). We show that the 2-OG/Fe(II)-dependent dioxygenase acts as methylguanidine hydroxylase (MgdH) that catalyzes the hydroxylation of the methyl groups of methylguanidine and N,N-dimethylguanidine, producing guanidine and formaldehyde after

spontaneous but slow decay of the hydroxylated intermediates. The N-(hydroxymethyl)guanidine product is unexpectedly stable in comparison to other hemiaminals produced in demethylation reactions[22]. We find that the hypothetical protein that accompanies the MgdH catalyzes the fragmentation of N-(hydroxymethyl)guanidine into formaldehyde and guanidine, acting as N-(hydroxymethyl)guanidine lyase (MgdL). *V. boliviensis* is able to utilize either guanidine or methylguanidine, but not N,N-dimethylguanidine, as sole nitrogen sources, and comparative proteome analysis showed increased expression of the entire *gca/mgdH*-containing operon in cells grown on methylguanidine or guanidine compared to those grown on ammonium.

## Results

### Genomic context of *mgdH*

The genes required for guanidine degradation via carboxy-guanidine are typically organized as a common operon and these operons are frequently regulated by guanidine riboswitches[2,9,10]. In the genome of *V. boliviensis*, we identified two additional genes in such an operon, encoding for a putative 2-OG/Fe(II)-dependent dioxygenase and a hypothetical protein (labelled MgdH and MgdL in Fig. 1). Similarly expanded guanidine degradation operons can be identified in other halophilic bacteria of the genera *Salinisphaera* and *Vreelandella*[2]. Sequence analyses did not reveal a clear substrate candidate for MgdH or any hint on the enzymatic function of MgdL. The operon is apparently regulated by two guanidine riboswitches located in the 5'-UTR of the operon as well as within the operon between *cgdB* and *gca*.

We utilized in-line probing[23] to confirm that the class I guanidine riboswitch (Gd-I) that is found upstream of the operon specifically binds guanidine with an apparent $K_D$ of $44 \pm 9\,\mu M$ (Fig. 2A, C) and is not binding methylguanidine, the substrate of MgdH described below, in the concentration range tested (0.005–1 mM). The second riboswitch, a class II guanidine riboswitch (Gd-II) residing in the intergenic region between *cgdB* and *gca*, remarkably did not discriminate between guanidine and methylguanidine with apparent $K_D$s of $93 \pm 6\,\mu M$ and $83 \pm 5\,\mu M$, respectively (Fig. 2B, D). In contrast, N,N-dimethylguanidine that was shown to be an alternative substrate for MgdH (see below), did not induce conformational changes with the more promiscuous Gd-II riboswitch (Supplementary Fig. 2). The Gd-I riboswitch presumably acts at the transcriptional level by destabilization of an intrinsic terminator, whereas the Gd-II riboswitch is proposed to act on translation by un-masking of the Shine–Dalgarno sequence in front of *gca*[2,4,24]. However, it has been shown that regulation of mRNA levels and translational control can be tightly linked in riboswitch regulated genes[25,26].

### MgdH is a methylguanidine hydroxylase

To determine the catalytic activity of the 2-OG/Fe(II)-dependent dioxygenase MdgH, we heterologously expressed *mgdH* in *Escherichia coli* BL21 (DE3) and purified the 6x His-tagged protein by nickel-affinity chromatography (Fig. 3A). In light of the conjunction with guanidine-metabolizing genes and guanidine riboswitches, we tested a collection of (methyl-)guanidine derivatives as potential substrates (Fig. 3B and Supplementary Table 1). With 2-OG as co-substrate, MgdH-dependent oxygen consumption was exclusively observed with methylguanidine and N,N-dimethylguanidine as substrates. Analysis of steady-state kinetic parameters yielded an apparent $K_M$ of $0.16 \pm 0.01\,mM$ and a $k_{cat}$ of $19.6 \pm 0.4\,s^{-1}$ for methylguanidine (concentration of 2-OG fixed at 10 mM, Fig. 3D) and an apparent $K_M$ of $0.53 \pm 0.05\,mM$ and a $k_{cat}$ of $20.5 \pm 0.5\,s^{-1}$ for 2-OG (concentration of methylguanidine fixed at 2.5 mM, Fig. 3E). For N,N-dimethylguanidine a $K_M$ $0.25 \pm 0.01\,mM$ of and a $k_{cat}$ of $17.3 \pm 0.2\,s^{-1}$ was determined (concentration of 2-OG was 5 mM, Fig. 3F). Dicarboxylic acids other than 2-OG did not stimulate detectable $O_2$ consumption with methylguanidine or N,N-dimethylguanidine as main substrates (Supplementary Table 1). Other members

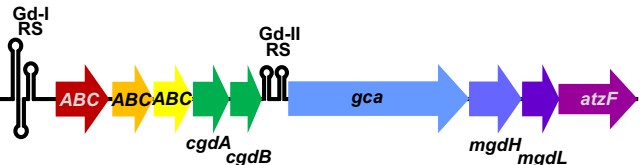

**Fig. 1 | Genomic arrangement of the guanidine degradation operon in *V. boliviensis*.** The operon is preceded by a class I guanidine riboswitch (Gd-I RS) and contains an additional internal class II guanidine riboswitch (Gd-II RS). The operon comprises the genes of the guanidine carboxylase pathway: guanidine carboxylase (*gca*), carboxyguanidine hydrolase subunits A and B (*cgdA/B*), and allophanate hydrolase (*atzF*). The three ABC genes encode for an ABC-transporter that likely functions as importer as it contains a periplasmic substrate binding subunit. The additional genes *mgdH* and *mgdL* are located between *gca* and *atzF*.

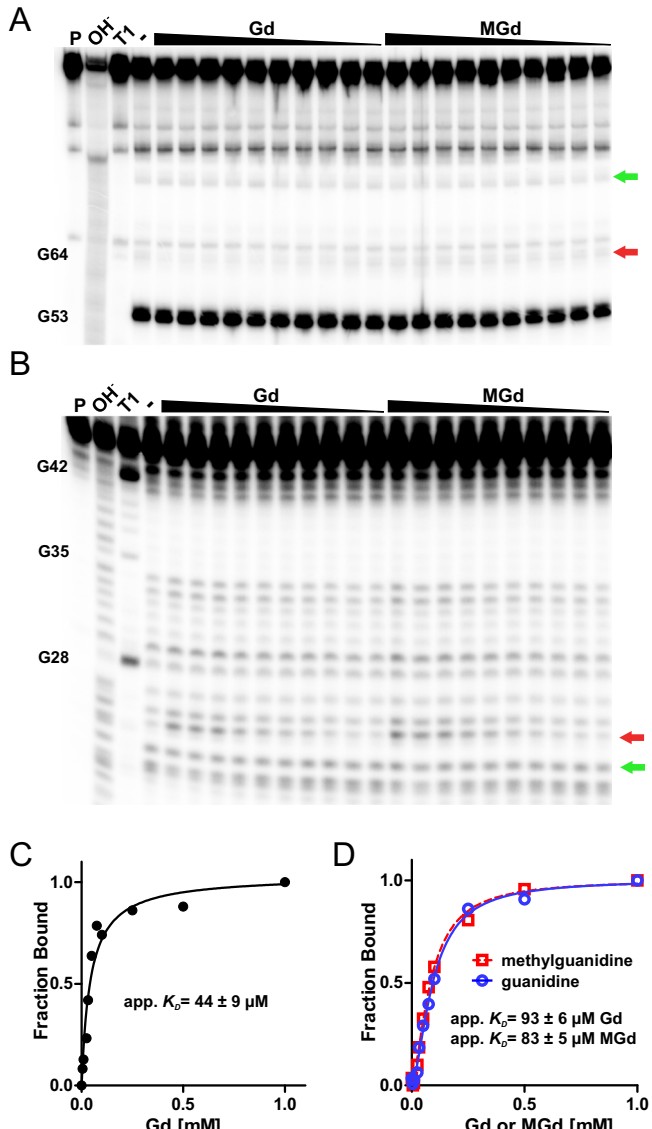

**Fig. 2 | Ligand affinity and specificity of the riboswitches in the guanidine carboxylase operon of *V. boliviensis*. A**, **B** Autoradiographs of in-line probing reactions to characterize the Gd-I riboswitch upstream of the operon (**A**) and the internal Gd-II riboswitch (**B**). Modulation of bands in dependence of ligand concentration indicates a binding event and or structural rearrangement of the RNA[23]. The fragments showing ligand-dependent changes in abundance that were used for $K_D$ determination are indicated by red arrows. The change in band intensity in (**A**) is poorly visible by eye and can best be seen by comparing the control lane (−) to the adjacent lane, where the sample with the highest guanidine concentration was loaded. The invariable fragments that were used for normalization of the signal intensity are indicated by green arrows. Precursor RNA (P) and reactions without ligand (−) were included on the gel as controls, as well as an alkaline digest (OH⁻) of the RNA and a digest with RNAse T1 that cleaves specifically at guanosines (T1) for nucleotide assignment. For pictures of the whole radiographs and results with N,N-dimethylguanidine see Supplementary Figs. 1 and 2. **C**, **D** Normalized intensities of modulated bands plotted against ligand concentrations. Apparent $K_D$ values were obtained by non-linear regression. Data points represent single measurements. The error indicates the standard error derived from the fit. R² = 0.957 (**C**); R² = 0.995 and R² = 0.996 for guanidine and methylguanidine, respectively (**D**). Consistent results were obtained in two independent experiments.

of the 2-OG/Fe(II)-dependent dioxygenase family are known to catalyze demethylation reactions of methylated nucleobases and amino acids by hydroxylation of the methyl group[20]. The resulting hydroxymethyl intermediates decay spontaneously, yielding formaldehyde

and the demethylated nucleobase or amino acid. We hypothesized that a similar reaction is catalyzed by MgdH. Therefore, the reaction was quenched after 1 h with methanol containing isotopically labelled guanidine as internal standard and subjected to LC-MS analysis to determine the concentration of guanidine. From the same reaction, we measured the concentration of formaldehyde colorimetrically with Nash reagent. We detected guanidine and formaldehyde in an approximately 1:1 ratio (Fig. 3C).

### Fragmentation of N-(hydroxymethyl)guanidine

When we analyzed the substrate consumption and product formation over time by LC-MS analysis, we observed an unexpected delay in product formation, which prompted us to search for alternative products of the reaction of MdgH with methylguanidine. We observed a single substance with an *m/z* ratio of 90 that accumulated transiently. The relative signal intensities of this substance could explain the difference between methylguanidine disappearance and guanidine formation, and the *m/z* ratio of 90 matches the expected reaction intermediate N-(hydroxymethyl)guanidine. We observed that the intermediate was more stable than we expected with an approximate half-life >1 h (Fig. 4A). We speculated that the hypothetical protein following *mgdH* might serve to accelerate the release of guanidine, giving rise to the tentative name MgdL. We heterologously expressed *mgdL* in *E. coli* BL21 (DE3), purified the protein by nickel-affinity chromatography (Supplementary Fig. 3) and included it into the MgdH assay (Fig. 4B). As with MgdH alone, methylguanidine was rapidly consumed. In contrast, we did not observe substantial levels of N-(hydroxymethyl)guanidine in the reaction containing both MgdH and MgdL. The fast formation of guanidine, simultaneous with the consumption of methylguanidine, indicated that in the coupled reaction the reaction of MgdH was rate-limiting. Comparing the reactions in presence and absence of MgdL, its presence clearly accelerated the fragmentation of N-(hydroxymethyl)guanidine, thus catalyzing the second step of methylguanidine demethylation following the initial hydroxylation by MgdH (Fig. 4C).

We built a substrate-bound atomic model of MgdL with N-(hydroxymethyl)guanidine using Chai-1[27] and used the Consurf[28] server to assess the conservation of potentially functional residues. Homologues were retrieved by HMMER from the Uniprot90 database with an HMMER *e*-value of ≤ 0.001 with a minimal sequence identity of 30%. A multiple sequence alignment of 150 sequences sampling the list of all hits (216 total) was built using MAFFT (Supplementary Data 1). The sequence comparison identified several highly conserved residues that are located inside a beta-barrel of the model (Supplementary Fig. 4A, C). The model suggests that the guanidine moiety of the substrate is bound by hydrogen bonding to glutamate and aspartate and additionally by cation-π stacking with a tryptophan. Strikingly, the substrate is positioned in a way that a conserved aspartate (D49) and arginine (R51) could accelerate the fragmentation by acid-base catalysis (Supplementary Fig. 4B). In the proposed mechanism, D49 would act as general base catalyst, while R51 would act as general acid catalyst (Fig. 4D). Although the side chain of arginine is commonly believed to not participate in acid-base catalysis due to the high p$K_a$ value of the guanidinium group, it has been shown before to serve as general acid catalyst in a serine recombinase[29]. Also, unprotonated guanidine is transiently formed during the reaction after cleavage of the C-N bond. As guanidine is a very strong base, it seems plausible that the subsequent protonation is mediated by the guanidinium group of an adjacent arginine residue. To test whether the conserved aspartate (D49) and arginine (R51) residues are important for catalysis, MgdL variants D49A and R51A, as well as the more conservative mutations D49E and R51K, were generated and purified (Supplementary Fig. 5A). Coupled reactions with MgdH and MgdL were set up as before and substrate consumption as well as product formation was monitored by LC-MS. In contrast to the reaction with wildtype MgdL, substantial and

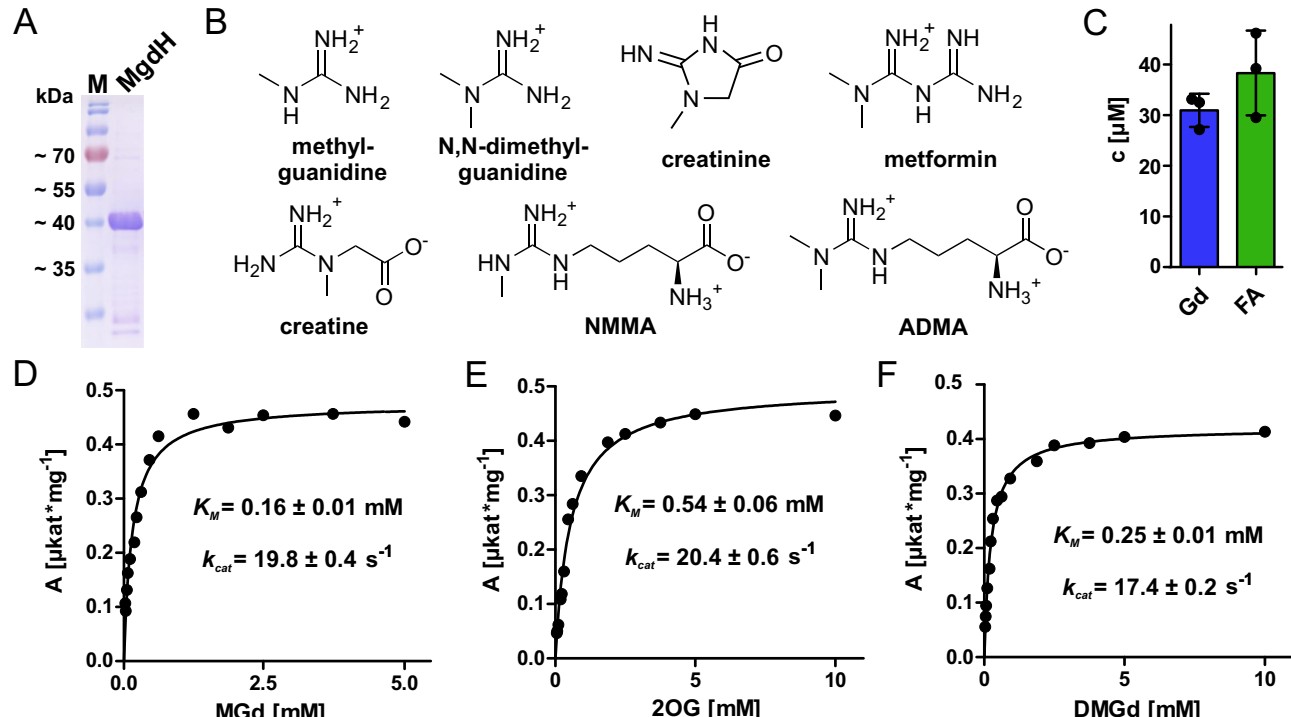

**Fig. 3 | Reaction and characterization of MgdH. A** Coomassie stained denaturing polyacrylamide gel of purified MgdH matching the expected mass of 41 kDa. Similar results were obtained for more than three independent purifications. **B** Selection of compounds tested as substrates. For a full list, see Supplementary Table 1. Only methylguanidine and N,N-dimethylguanidine induced $O_2$ consumption by MgdH. $N^G$-monomethyl-L-arginine (NMMA); asymmetric dimethylarginine (ADMA). **C** Products formed after 1 h in the MgdH reaction with methylguanidine. Guanidine (Gd) was measured via LC-MS with isotopically labelled guanidine as internal standard and formaldehyde (FA) was colorimetrically measured with Nash reagent. Columns represent the average of three independent enzyme reactions ($n = 3$; error bars, s.d.). **D–F** Specific activities of oxygen consumption at various co-substrate and substrate concentrations were analyzed with a Clark-type oxygen electrode. The concentrations of either the co-substrate 2-oxoglutarate or the substrates methylguanidine (MGd) and dimethylguanidine (DMGd) were fixed at 10 mM or 2.5 mM, respectively. Specific activities were calculated from the slope of the initial linear reaction rate. Apparent $K_M$ and $A_{max}$ were determined by non-linear regression using the Michaelis–Menten equation and $k_{cat}$ was calculated subsequently. Standard errors were calculated from the regression curves ($R^2 = 0.9825$ (**D**), $R^2 = 0.9830$ (**E**), and $R^2 = 0.9948$ (**F**)). Data points represent single measurements, consistent results were obtained with two independent enzyme preparations.

prolonged accumulation of N-(hydroxymethyl)guanidine was observed with all MgdL variants in a similar manner as in the absence of MgdL (Fig. 4E, F and Supplementary Fig. 5B). To assess whether MgdL variants have residual N-(hydroxymethly)guanidine fragmentation activity, we determined the concentration of guanidine after 40 min. Note that regardless of the presence or absence of any MgdL variant, virtually all methylguanidine was consumed within 10 min, and simultaneously converted to guanidine when wildtype MgdL was present. In the reactions with the MgdL variants, guanidine levels after 40 min were similar to the control reaction without MgdL confirming the importance of D49 and R51 for the functionality of MgdL and thus supporting the proposed catalytic mechanism (Fig. 4G).

When we used N,N-dimethylguanidine as substrate for MgdH, we observed both the mono- and di-hydroxylated N,N-dimethylguanidine intermediates (Supplementary Fig. 6A). Unlike N-(hydroxymethyl)guanidine, the levels of hydroxylated N,N-dimethylguanidine species were insensitive to the presence of MgdL (Supplementary Fig. 6B). Consistent with the lower $K_M$ value of the MdgH reaction with methylguanidine compared to N,N-dimethylguanidine, methylguanidine was only detected at low levels, as it was presumably directly hydroxylated when formed. In accordance with the results obtained with methylguanidine as substrate, substantial amounts of N-(hydroxymethyl)guanidine were only measured in the absence of MgdL (Supplementary Fig. 6A). We propose that the reaction can proceed via the spontaneous fragmentation of N-hydroxymethyl-N-methylguanidine to methylguanidine or that a second hydroxylation to N,N-di(hydroxymethyl)guanidine can take

place that decays spontaneously to N-(hydroxymethyl)guanidine (Supplementary Fig. 6C). Thus, in contrast to MgdH that apparently accepted even N-hydroxymethyl-N-methylguanidine as substrate, MgdL exhibited a narrow substrate specificity for N-(hydroxymethyl) guanidine. The results suggest that methylguanidine is the natural substrate of the demethylase system. This notion is supported by the specificity of the Gd-II riboswitch that is only induced by guanidine and methylguanidine (Fig. 1 and Supplementary Fig. 2).

**Methylguanidine as nitrogen source for *V. boliviensis***

In order to assess the relevance of the presence of methylguanidine-degrading enzymes encoded in the genome of *V. boliviensis*, we tested if these bacteria can utilize methylguanidine as sole nitrogen source. *V. boliviensis* grew readily on 5 mM guanidine or methylguanidine as nitrogen sources (Fig. 5). We repeated the experiment with 1 mM substrates and monitored the methylguanidine and guanidine concentrations in the actively growing culture. With increasing cell densities, the levels of methylguanidine and guanidine decreased (Supplementary Fig. 7). The presence of only atmospheric $N_2$ did not stimulate any growth of *V. boliviensis* (Supplementary Fig. 7A). Unexpectedly, N,N-dimethylguanidine as sole nitrogen source did not support rapid growth of *V. boliviensis* and only residual growth was observed after 35 h (Fig. 5A Supplementary Fig. 7B). A potential reason for the observed growth deficiency is that dimethylguanidine is not a ligand for the Gd-II riboswitch and thus cannot enhance expression of *mgdH* and *mgdL* (Supplementary Fig. 2). Additionally, MgdL seemed not to accelerate the fragmentation of N-(hydroxymethyl)-N-methyl-

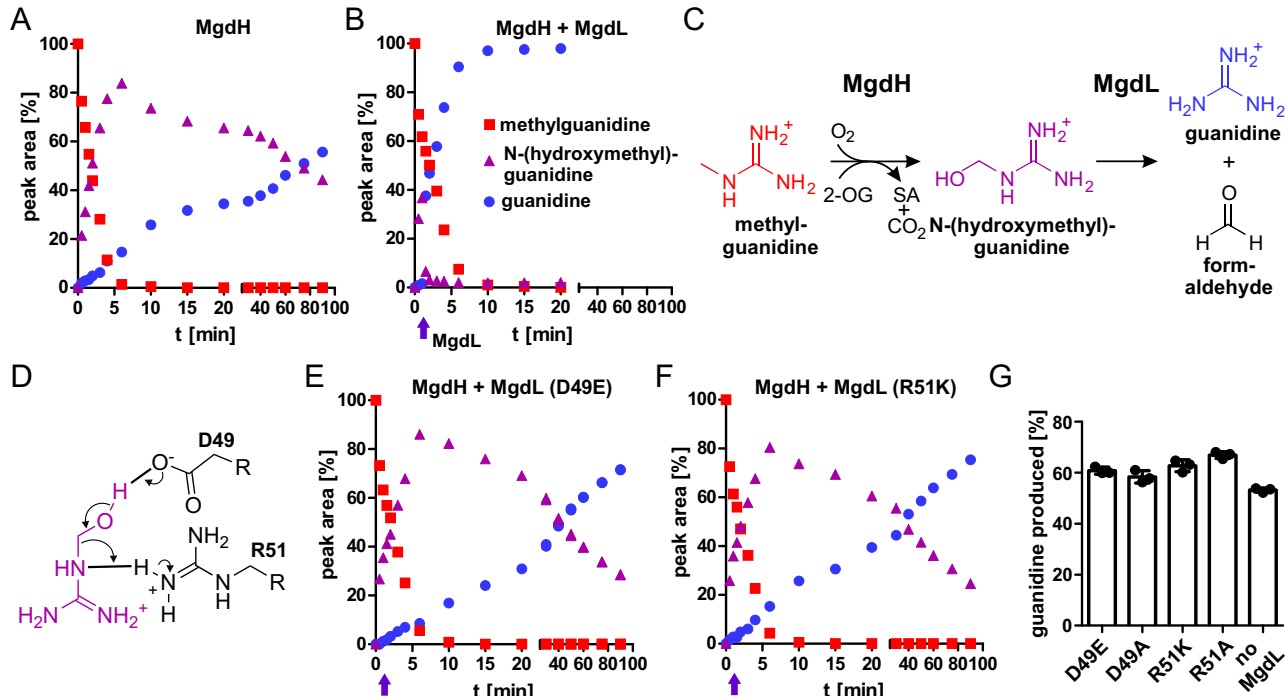

**Fig. 4 | LC-MS analysis of the enzymatic demethylation of methylguanidine and proposed reaction mechanism. A** Methylguanidine (0.5 mM) was incubated with 5 mM 2-OG and 10 μg/mL MgdH. Consumption of methylguanidine (red squares) and generation of N-(hydroxymethyl)guanidine (purple triangles) and guanidine (blue dots) were monitored by LC-MS. 100% peak area was defined as the peak area of 0.5 mM methylguanidine or 0.5 mM guanidine. N-(hydroxymethyl)guanidine % peak area was calculated as the difference of the combined methylguanidine and guanidine % peak areas to 100%. **B** The reaction was set up as described in (**A**), and 20 μg/mL MgdL were added after 1.5 min as indicated by the purple arrow. In reactions with wildtype MgdL, N-(hydroxymethyl)guanidine was rapidly fragmented and complete turnover to guanidine was observed simultaneous to the consumption of methylguanidine. Data represent single data points. Consistent results were obtained with independent enzyme preparations. **C** Scheme of the demethylation reaction of methylguanidine with N-(hydroxymethyl)guanidine as intermediate to guanidine as monitored in (**A** and **B**). The co-substrate 2-oxoglutarate (2-OG) is converted to succinate (SA). **D** Proposed reaction mechanism. The

decay of N-(hydroxymethyl)-guanidine could be accelerated via acid-base catalysis by MgdL. A conserved aspartate (D49) is positioned to act as the general base that abstracts a proton from the hydroxyl group of N-(hydroxymethyl)-guanidine. The neighboring conserved arginine (R51) could function as a general acid, donating a proton from its guanidinium group to the bridging nitrogen of the substrate. **E, F** The activity of MgdL variant D49E (**E**) and R49K (**F**) was performed as described in (**B**). Data represent single data points. Consistent results were obtained with independent enzyme preparations. **G** Residual activity of MgdL variants: Reactions were set up as described in (**B**) with MgdL variants as annotated. Produced guanidine was determined by LC-MS after 40 min as a measure for the fragmentation of N-(hydroxymethyl)-guanidine. In the reactions with MgdL variants, N-(hydroxymethyl)-guanidine was formed and decayed to guanidine with similar kinetics as in the control reaction without MgdL (no MgdL). Columns represent the average of three independent enzyme reactions and consistent results were obtained with independent preparations ($n = 3$; error bars, s.d.).

guanidine and N,N-di(hydroxymethyl)-guanidine (Supplementary Fig. 6). Furthermore, dimethylguanidine might not be efficiently taken up by *V. boliviensis* if the substrate binding protein of the ABC transporter is selective for guanidine and methylguanidine. *V. boliviensis* was not able to grow in minimal medium containing methylguanidine but no additional carbon and energy source (data not shown). A proteome analysis was conducted to evaluate the role of the *gca/mgdH* operon in the growth of *V. boliviensis* in three different conditions: Utilizing either methylguanidine, guanidine, or ammonium as sole nitrogen sources. When grown on methylguanidine and guanidine, all proteins encoded by the guanidine carboxylase operon were highly upregulated. Most proteins were not detected when *V. boliviensis* was grown on ammonium, with the exception of guanidine carboxylase, allophanate hydrolase, and the ATP-binding protein of the putative transporter albeit at levels approximately three orders of magnitude lower compared to the (methyl-)guanidine samples (Fig. 5B and Supplementary Data 2). The formaldehyde released during the degradation of methylguanidine could be toxic to the cells. We did not detect formaldehyde in *V. boliviensis* cultures growing actively with methylguanidine as sole nitrogen source (data not shown, detection limit ≤10 μM, Supplementary Fig. 9). Instead, we found that a probable formaldehyde detoxification system comprising predicted S-(hydroxymethyl)glutathione dehydrogenase (NCBI Acc.No.: KUC_2684) and

S-formylglutathione hydrolase (NCBI Acc.No.: KUC_2683)[30] was upregulated when *V. boliviensis* grew on methylguanidine (Supplementary Data 2).

## Discussion

We show that the halophilic bacterium *V. boliviensis* can use methylguanidine as sole source of nitrogen with the activities of two enzymes. The bacteria possess a guanidine carboxylase pathway for the degradation of guanidine[2,9,10] that is expanded by the 2-OG/Fe(II)-dependent dioxygenase MgdH and a lyase MgdL. These two enzymes jointly catalyze the demethylation of methylguanidine and the concomitant release of formaldehyde. While MgdH catalyzes the hydroxylation of methylguanidine to the surprisingly long-lived hemiaminal N-(hydroxymethyl)guanidine, MgdL accelerates the decay of N-(hydroxymethyl)guanidine to guanidine and formaldehyde. Expression of the entire guanidine carboxylase operon was highly upregulated when cells were grown with guanidine or methylguanidine as sole nitrogen source. This upregulation is apparently mediated by the combined action of two riboswitches: A Gd-I riboswitch presumably acting on transcription[2] resides in the 5′-UTR of the operon and specifically binds guanidine while it discriminates against methylguanidine. The second riboswitch is a Gd-II riboswitch exerting presumably translational control[4] and is located upstream of *gca*. This Gd-II riboswitch responds

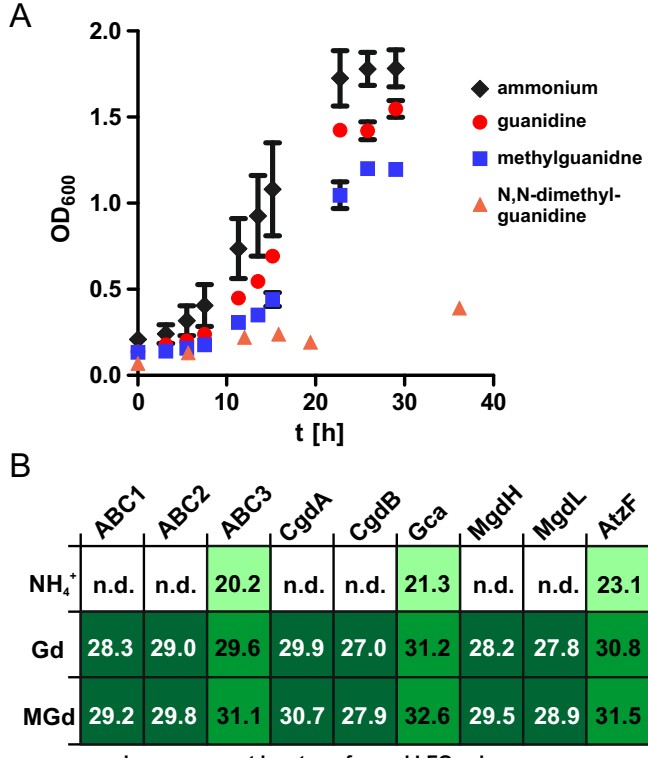

**A**

**B**

| | ABC1 | ABC2 | ABC3 | CgdA | CgdB | Gca | MgdH | MgdL | AtzF |
|---|---|---|---|---|---|---|---|---|---|
| NH₄⁺ | n.d. | n.d. | 20.2 | n.d. | n.d. | 21.3 | n.d. | n.d. | 23.1 |
| Gd | 28.3 | 29.0 | 29.6 | 29.9 | 27.0 | 31.2 | 28.2 | 27.8 | 30.8 |
| MGd | 29.2 | 29.8 | 31.1 | 30.7 | 27.9 | 32.6 | 29.5 | 28.9 | 31.5 |

**numbers represent log₂ transformed LFQ values**

**Fig. 5 | Methylguanidine as nitrogen source for *V. boliviensis*. A** *V. boliviensis* was grown in minimal medium with glucose as carbon and energy source. The medium was supplemented with either 5 mM guanidine (red dots), 5 mM methylguanidine (blue squares), 5 mM N,N-dimethylguanidine (orange triangles) or 15 mM ammonium chloride (black diamonds) as nitrogen source. OD₆₀₀ was measured in culture tubes. Data points represent the mean of three independent cultures and error bars indicate standard deviation. (*n* = 3; error, s.d.) **B** Comparative proteome analysis of *V. boliviensis* grown in medium with ammonium (NH₄⁺), guanidine (Gd), or methylguanidine (MGd) as sole N-source. LFQ intensity values (log2) are given for the proteins encoded by the guanidine carboxylase operon for the respective condition. Dark green background indicates proteins that were detected only when *V. boliviensis* grew with guanidine or methylguanidine, whereas lighter green backgrounds indicate proteins that were much more abundant under these conditions compared to the ammonium control. (n.d. not detected; Supplementary Data 2).

with the same sensitivity to both methylguanidine and guanidine. The control by two types of guanidine-responsive riboswitches with different substrate specificities and acting on transcription and translation could be an interesting subject for further studies.

With the present study, two enzyme activities are described: One type of 2-OG/Fe(II)-dependent dioxygenases and a previously unknown lyase activity. The dioxygenase has some functional similarity to demethylases utilized to remove epigenetic marks and post-translational methylations found on nucleobases and amino acid side chains[20,31]. In case of such erasers of nucleic acid or protein methylation modifications, it is believed that the intermediate hemiaminal, produced by the hydroxylation of the methyl group, is spontaneously decaying into the demethylated product and formaldehyde[22]. Although the hemiaminal intermediate has been observed experimentally before[21], the stability of the N-(hydroxymethyl)guanidine is to our knowledge unprecedented for a hemiaminal occurring in biological systems. Guanidine with its delocalized electrons[32] represents an electron-poor system that probably stabilizes the intermediate hemiaminal. We are only aware of one activity similar to the described lyase of MdgL where the decay of a hydroxylated intermediate is accelerated

by a dedicated enzyme: Certain peptide hormones are stabilized by formation of a C-terminal carboxamide by the action of a Cu²⁺-dependent peptide-amidating monooxygenase[33]. The initially formed α-hydroxylated glycine residue is subsequently cleaved by a peptidyl-α-hydroxyglycine-α-amidating lyase (PAL)[34,35]. However, although a similar reaction is catalyzed, PAL and MdgL do not share sequence homology.

The $K_M$ of the MdgH reaction for 2-OG falls into the range (~ 0.3 to 12 mM in *E. coli*) of 2-OG concentrations expected in bacteria[36]. As comprehensive and quantitative data for methylated guanidine concentrations in nature are lacking, it is difficult to speculate if the reported $K_M$ values support biological relevance. If the ABC-importer encoded within the *mgdH*-containing operon accepts both guanidine and methylguanidine as substrates, they could be accumulated in the cells to concentrations far above those in the environment. Although there are no studies addressing potential guanidine importers, it has been shown that guanidine exporters also accept methylguanidine as substrate[5]. Potential sources of methylguanidine in natural systems remain elusive, except for studies reporting abiotic production from creatinine[37,38]. *V. boliviensis* was isolated from soil of a salt lake at high altitude[39] where harsh conditions could result in the accumulation of methylguanidine. Given that creatinine is a metabolic waste product in all vertebrates, the amount of methylguanidine produced by abiotic decay may be sufficient to support the evolution of an enzyme system for uptake and subsequent degradation of methylguanidine via guanidine to ammonium for nitrogen assimilation. It is also conceivable that *V. boliviensis* or another organism produces methylguanidine enzymatically, but so far we are not aware of experimental evidence for such an activity. We have recently characterized MefH and its evolutionary precursor DmgH, which catalyze the hydrolysis of the type II anti-diabetic drug metformin and N,N-dimethylguanidine, respectively[40]. Although also in this case the natural source of N,N-dimethylguanidine remained elusive, our findings point to a so far underappreciated role of methylated guanidine species in nature.

## Methods

### Bacterial cultivation

*V. boliviensis* LC1 (DSM 15516) was obtained from the DSMZ and was grown routinely on Marine Broth (Difco) + 6% NaCl at 28 °C. Growth assessment of *V. boliviensis* on different nitrogen sources was performed in a high salt minimal medium (6% NaCl, 8.5 g/L Na₂HPO₄*2H₂O, 3 g/L KH₂PO₄, 0.5 g/L NaCl, 2 mM MgCl₂, 100 μM CaCl₂) that was supplemented with trace elements (0.1 mM EDTA, 0.03 mM FeCl₃, 6.2 μM ZnCl₂, 0.76 μM CuCl₂, 0.42 μM CoCl₂, 1.62 μM H₃BO₃; 0.08 μM MnCl₂) and vitamins (0.1 mg/L cyanocobalamin, 0.08 mg/L 4-aminobenzoic acid, 0.02 mg/L D-(+)-biotin, 0.2 mg/L niacin, 0.1 mg/L Ca-D-(+)-pantothenic acid, 0.3 mg/L pyridoxamine hydrochloride, 0.2 mg/L thiamine hochloride). The minimal medium contained 0.4% glucose as carbon source and 5 mM of either guanidine, methylguanidine, or N,N-dimethylguanidine or 15 mM of ammonium chloride. To test if *V. boliviensis* grows on methylguanidine without additional carbon source, glucose was omitted from the medium. Bacterial growth was assessed by measuring the OD₆₀₀ in a test tube photometer or in disposable micro-cuvettes. *E. coli* SoluBL21 (Genlantis) or *E. coli* BL21 (λDE3) gold (Invitrogen) were used for recombinant protein production and were grown at 37 °C in LB with 50 μg/ml kanamycin.

### Cloning, protein overexpression and purification

*mgdH* and *mgdL* were amplified from *V. boliviensis* genomic DNA with forward primers 5′-CCTGTACTTTCAAGGT-GCTATGAGCCAAGCATC-CAACGAC and 5′-CCTGTACTTTCAAGGTG-CTATGAGCGTTTCCTTAC CC and reverse primers 5′-GGTGGTGCTCGAGTGCATTATGGGGTTTC-CTTATGCTTGC and 5′-GGTGGTGCTCGAGTGCTCAAGTGATTTGATT-GAAAG, respectively. The PCR products were cloned into a pET24

derivative by Gibson assembly to obtain expression constructs for enzymes with an N-terminal 6 × His-tag followed by a TEV cleavage site. For recombinant protein expression, *E. coli* SoluBL21 (Genlantis) or *E. coli* BL21 (λDE3) gold (Invitrogen) transformed with the expression constructs was grown at 37 °C to an $OD_{600}$ of 0.6, transferred to 18 °C and induced over night with 1 mM IPTG. The cells were harvested by centrifugation, resuspended in enzyme buffer (50 mM $Na_2HPO_4$ pH 8, 300 mM NaCl) supplemented with 1× EDTA-free cOmplete protease inhibitor (Roche) and lysed by ultrasonication (Branson). After centrifugation at $12000 \times g$ for 20 min at 4 °C, the soluble protein fraction was incubated with HisBind NiNTA-agarose resin (Qiagen, Hilden, Germany) for 30 min at 4 °C. The resin was loaded into gravity flow columns and washed sequentially with enzyme buffer supplemented with 20 mM or 50 mM imidazole. The His-tagged enzymes were eluted in enzyme buffer supplemented with 500 mM imidazole and desalted into enzyme buffer by passage through PD10 MidiTrap columns (GE Lifesciences/Cytiva).

## Oxygen consumption measurements

Enzymatic activity was monitored via oxygen consumption in an Oxygraph+ Clark-type oxygen electrode with the OxyTrace+ software (Hansatech). The instrument was calibrated with air-saturated and oxygen-free water. For enzyme reactions, 2 μg of enzyme were incubated at 30 °C in 1 mL of 80 mM KCl, 20 mM NaCl, 2 mM $MgCl_2$, 40 μM $(NH_4)_2Fe(II)SO_4$, 400 μM ascorbate, and 100 mM MOPS pH 7.2 with substrate and 2-OG. In the substrate screen, the co-substrate 2-OG was 5 mM and substrates were 10 mM. For the establishment of the Michaelis–Menten kinetics, either 2-OG or methylguanidine was fixed at saturating concentrations, 5 mM or 2.5 mM, respectively, and the concentration for the analyzed compound was varied from 50 μM to 10 mM for 2-OG, 25 μM to 5 mM for MG or 50 μM to 10 mM for N,N-dimethylguanidine. Specific activities were plotted against the concentration and the data were fitted to the Michaelis–Menten equation with GraphPad Prism.

## LC-MS analysis

Guanidine, methylguanidine and N,N-dimethylguanidine concentrations were measured by high-performance liquid chromatography coupled with mass spectrometry on a Prominence HPLC system with LCMS-2020 single quadrupole MS (Shimadzu) as described previously[41]. Aliquots of the enzymatic reactions (10 μL) were quenched with 40 μL methanol containing 10 μM $^{13}C^{15}N$-guanidine (Sigma-Aldrich 607312) and subjected to LC-MS. The isotopically labeled standard was used as reference to quantify guanidine. 2 μL sample were injected into a Nucleodur HILIC column (250 mm × 2 mm, 3 μm particle size, Macherey-Nagel). The mobile phases contained 10 mM ammonium formate and 0.2% formic acid in water (eluent A) and 90% acetonitrile, 10 mM ammonium formate and 0.2% formic acid (eluent B). Compounds were eluted with a gradient from 90% to 60% of eluent B in 6 min, than to 45% of eluent B in 1 min, followed by a isocratic step of 45% eluent B for 3 min at a flow rate of 0.15 ml min⁻¹. The separated compounds were analyzed by single ion monitoring in positive mode of the $m/z$ values corresponding to the predicted protonated ions of guanidine, methylguanidine, N,N,-dimethylguanidine and their hydroxylated products. LC-MS chromatograms were evaluated with LabSolutions v. 5.72 (Shimadzu) and the peak area was taken as measure for the respective compound concentration (Supplementary Fig. 8). Calibration factors for methylguanidine and N,N-dimethylguanidine were obtained from standard solutions within the measured concentration ranges. Calibration factors for the hydroxylated intermediates were estimated based on the predicted reaction stoichiometry. For the time-resolved LC-MS analysis of the enzyme reactions 0.5 mM methylguanidine were incubated with 5 mM 2-OG and 10 μg/mL MgdH. MgdL (20 μg/mL) was added after 1.5 min in the respective experiments. Aliquots of the reaction were taken at different time

points, quenched in methanol as described and immediately subjected to LC-MS. Compounds were eluted isocratically with 45% (v/v) acetonitrile, 10 mM ammonium formate, 0.2% (v/v) formic acid in water at a flow rate of 0.2 ml min⁻¹. 100% peak area was defined as the peak area of 0.5 mM methylguanidine or 0.5 mM guanidine. The % peak areas of hydroxylated methyl- and N,N-dimethylguanidine were calculated as the difference of the combined methylguanidine, N,N-dimethylguanidine and guanidine % peak areas to 100%.

## Detection of formaldehyde by Nash reaction

50 μl of an MgdH assay under optimal conditions were withdrawn from the sealed Clark oxygen electrode after complete consumption of oxygen. The reaction was mixed with 150 μl Nash's reagent (2 M ammonium acetate, 50 mM acetic acid, 20 mM acetylacetone)[42], diluted with 100 μl water and immediately incubated at 60 °C for 5 min to stop the enzyme reaction and develop the color. For the detection of formaldehyde in actively growing cultures of *V. boliviensis* on methylguanidine 150 μl of the culture were mixed with 150 μl of Nash's reagent. As positive control culture samples were spiked with 25 μM formaldehyde. Absorbance was measured in a 96-well microtiter plate with the Tecan Infinity plate reader at 412 nm. Samples without enzyme or substrate were measured as negative controls and a freshly prepared formaldehyde concentration series was used to establish a standard curve (Supplementary Fig. 9).

## In-line probing assay

Generation of 5′-end-labeled RNA and subsequent in-line probing was performed as described by Breaker and Soukup[23]. DNA templates containing the riboswitch sequence were generated by PCR with a forward primer containing the T7 promoter at its 5′-end (Supplementary Table 2). Purified PCR products were used for in vitro transcription with T7 RNA polymerase. RNase Inhibitor and PPase were added for more efficient transcription. Full-length RNA was purified by polyacrylamide gel electrophoresis. A 20 pmol aliquot of RNA was dephosphorylated with recombinant shrimp alkaline phosphatase and $^{32}P$-labeling was achieved by T4 polynucleotide kinase with γ-[$^{32}P$]-ATP (Hartman Analytics). Radiolabeled RNA was purified by polyacrylamide gel electrophoresis. In-line probing reactions were set up with approximately 1 kBq of labelled RNA and the respective ligand concentration in 50 mM Tris-HCl pH 8.3, 20 mM $MgCl_2$, 100 mM KCl at 25 °C. The RNA was incubated for 40 h at 25 °C. For nucleotide assignment, partial digestion of the RNA with RNase T1 and alkaline digestion were performed as described previously[43]. Controls and in-line probing reactions were analyzed by polyacrylamide gel electrophoresis and imaged using a Typhoon FLA 7000 phosphorimager. Band intensities of modulated and unmodulated bands were determined using Quantity One (Bio-Rad). Modulated band intensity was referenced against the unmodulated band intensity of the respective lane. Intensities were normalized to the lowest and highest values and plotted against the ligand concentration. Data were fitted with nonlinear regression assuming one binding site for Gd-I[2,44] and two binding sites for Gd-II[4,45]. Thus, for Gd-II the Hill-equation was used for regression.

## Proteome Analysis

*V. boliviensis* was grown in high salt minimal media for about 35 to 40 h at 28 °C and 210 rpm until reaching an $OD_{600}$ of about 1 to 1.5. Cells were harvested by centrifugation at 4 °C. Cells were lysed by sonication with a Branson Sonifier in 1× PBS + 1% (w/v) SDS. Total protein amount was determined with the BCA method (Thermo Scientific) according to the manufacturer's protocol. 45 μg total protein were loaded on a denaturing 14% polyacrylamide gel and electrophoresis was performed until the sample entered the resolving gel. The Coomassie-stained protein bands were cut out from the gel and sent for proteome analysis to the in-house proteomics facility.

## Reporting summary

Further information on research design is available in the Nature Portfolio Reporting Summary linked to this article.

## Data availability

The source data of all graphs and figures are provided in the Source Data. Semi-quantitative proteome data are provided in Supplementary Data 2. Source data are provided with this paper.

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

## Acknowledgements

We thank Astrid Joachimi and Dr. Dmitry Galetskiy for technical assistance. We thank the proteomics facility at the University of Konstanz.

## Author contributions

M.S., D.F., and J.S.H. conceived the project. F.G., C.K., C.B., M.S., and D.F. performed the experiments. M.S., D.F., and J.S.H. wrote the manuscript and prepared figures. The manuscript was reviewed and approved by all authors.

## Funding

## Competing interests

The authors declare no competing interests.
