## [Transparent Peer Review file · Nature Communications]

Demethylation of methylguanidine by a stepwise dioxygenase and lyase reaction

Corresponding Author: Professor Jörg Hartig

Version 0:

Reviewer comments:

Reviewer #1

(Remarks to the Author)

Reviewer #2

(Remarks to the Author)

This manuscript by Sinn et al. and Hartig describes the identification and validation of two novel enzymes for the oxidation of methylguanidine and the decomposition of the resulting product into guanidine and formaldehyde. The enzymes were identified by their association with two classes of guanidine-binding riboswitches, and this genomic analysis approach is proving to be a powerful way to link unusual proteins with their biochemical functions via riboswitch association. Their findings also expand the list of enzymes that manipulate guanidine or its close chemical derivatives, and help reveal previously unknown ways in which guanidine is involved in the metabolism and physiology of many species. The importance of this and related work is likely to increase as more discoveries are made.

The work appears to be carefully done, the results appear to be sound, and the manuscript is generally well written. I only have very minor comments that the authors might wish to consider when preparing their manuscript for publication. Importantly, none of these comments should be used to hold up publication.

Minor Comments

1. The title is a bit obscure and will be uninteresting to most readers. I recommend putting 'guanidine' in the title.
2. Line 37: Write "and" in place of "end".
3. Line 99: Flip references 23 and 24 so that the in-line probing method can be cited on its first mention in the text.
4. Fig. 1: Make gene names white when placed in a dark-colored arrow.
5. Line 101: Delete the extra space after "below".
6. Fig. 2A: Band changes are not clear on visual inspection, such that it is not clear if the arrows are in the right locations. The same is true for SI Figure 1. It would be good if the authors commented on the lack of obvious changes in band intensity.
7. Also in the gel figures, the authors use -OH and T1 nuclease to generate marker bands, but they don't use these to label the band locations to aid the reader. I don't think this causes doubt about the interpretations/conclusions, but it would be great if bands could be annotated.
8. Legend to Fig. 4, second to last line: Remove the dash from "Hill-equation".
9. Line 151: Remove the dash from "Nash-reagent".
10. Line 159: The "D/E/F" notation is different from other legends.
11. Line 182: Delete the space between "Chai-1" and "25".
12. Line 226 and elsewhere: Write "as a sole nitrogen source" in place of "as sole nitrogen source".
13. Line 278: Delete the extra space between "electrons" and "28".
14. References: Use the journal format for reference titles (for example, see the difference between Reference 1 and 2 regarding the use of upper-case letters).
15. SI Figure 5: I have a minor difference of opinion regarding the proposed reaction mechanism. If I were designing a fast

enzyme, I would have the positively charged nitrogen of R51 pointing a proton at the nitrogen atom bonded to the hydroxymethyl group. The positively charged nitrogen would more easily donate a proton than the adjacent NH₂ group.

Reviewer #3

(Remarks to the Author)

The manuscript describes the characterization of two new enzymes, MgdD and MgdL, whose genes were found in the guanidine degradation operon in *V. boliviensis*. MgdD was shown to be a 2-oxoglutarate/Fe(ii)-dependent dioxygenase that hydroxylates methylguanidine. The stable N-hydroxymethylguanidine is converted to guanidine and formaldehyde by the newly-discovered N-hydroxymethylguanidine lyase MgdL. The study is interesting and reveals the mechanism for an important biochemical reaction. The manuscript is well written, and the study is well developed. I did not notice any major flaws in methodology or data analysis. However, there are a few items that the authors should address:

1. Lines 67-70: The authors mention that the guanidine operon in *V. boliviensis* is similar to guanidine operons in other bacteria, but with two additional genes (*mgdD* and *mgdL*). How similar are the other genes in the operon to operons from other species? It would be helpful to perform a more in-depth bioinformatic analysis between the *V. boliviensis* operon and homologous operons from other species. Further, are the genes all in the same orientation? Organized in the same order? Are all of the other operons the same? The authors could consider including other operons in Figure 1 to compare with the operon in this study.
2. Figure 1: the *atzF* black text in the purple arrow is difficult to read.
3. Lines 106-108: The authors should demonstrate that the Gd-I riboswitch acts through transcriptional control and the Gd-II riboswitch acts through translational control.
4. Line 152: Should the callout for Figure 3C be Figure 2C?
5. Figure 2 legend: Please define DMGd and describe the reaction in the legend. I see a description for the graphs in parts D and E, but no information for part F.
6. SI Figure 4 legend (line 42): "E126 and D112 could form hydrogen bonds..." Should the "for" be "form"?

Version 1:

Reviewer comments:

Reviewer #1

(Remarks to the Author)

I have re-reviewed the manuscript by Sinn et al., now entitled "Demethylation of methylguanidine by a stepwise dioxygenase and lyase reaction" (previously: "Demethylation catalyzed by a step-wise dioxygenase and lyase tandem reaction").

The authors have addressed all of my previous comments and incorporated the additional data I requested in a satisfactory manner. The manuscript has improved significantly as a result of the revision.

However, I did notice a few minor typographical and/or formatting errors that should be corrected prior to publication.

Line 90: For consistency, "MgdD" should be changed to MgdH.

Line 113: "Fig. 2D" must be changed to 3D.

Line 114: "Fig. 2E" must be changed to 3E.

Line 116: "Fig. 2F" must be changed to 3F.

Line 125: "Fig. 2C" must be changed to 3C.

Line 145: "m/z" – must be italic.

Line 149: Which *E. coli* strain was used?

Lines 165 and 172: "D48" must be changed to D49.

Reviewer #2

(Remarks to the Author)

The authors have addressed all the original minor concerns I noted, and all looks good to proceed with publication.

Reviewer #3

(Remarks to the Author)

The authors have adequately addressed my concerns.

We thank all reviewers for the careful reading of our manuscript and constructive criticism and advice. We have carefully addressed all issues raised in the revised version of the manuscript.

Reviewer 1:

In this study, Sinn and colleagues uncover a novel guanidine metabolism pathway in *Vreelandella boliviensis*, regulated by guanidine-responsive riboswitches. Researchers identified two previously uncharacterized proteins within a riboswitch-controlled operon: (i) a 2-oxoglutarate/Fe(II)-dependent dioxygenase MgdD, which hydroxylates methylguanidine to N-hydroxymethylguanidine; and (ii) an N-(hydroxymethyl)guanidine lyase MgdL that accelerates its breakdown into guanidine and formaldehyde. The authors describe that *V. boliviensis* can use either guanidine or methylguanidine as its sole nitrogen source, claiming that these findings reveal two new enzymatic activities and expand the understanding of guanidine assimilation and regulation in halophilic bacteria. Although the results of this study, mainly resulting from activity assays with recombinant MgdD and MgdL, structural predictions, mass spectrometry and growth experiments using either ammonia or guanidine/methylguanidine/dimethylguanidine look interesting, additional in-depth analyses are needed to enhance the impact and quality of the publication. Moreover, there are several aspects that require clarification. These points will be addressed in the following sections.

Major review

Title:

The authors should consider revising the title, as it is currently too broad and lacks scientific precision.

We now changed the title to “Demethylation of methylguanidine by a stepwise dioxygenase and lyase reaction” that reflects our findings more precisely.

Abstract:

Line 22-23: This statement is not entirely accurate, as Hausinger et al. already reported the broad activity of 2-OG/Fe(II) dioxygenases in 2015 (doi: 10.1039/9781782621959-00001). The authors should consider softening this claim to reflect prior findings.

We were referring to the specific reaction of MgdD (now MdgH) and not the general capacity of 2-OG/Fe(II) dioxygenases to catalyse hydroxylation/demethylation reactions. For clarity we removed “so far unknown” and added the phrase “that jointly catalyze the demethylation of methylguanidine”

Results:

Line 82: The authors should consider renaming this subsection, as it primarily focuses on the characterization of the two riboswitches and their kinetics.

We changed the subsection title to “Genomic context of *mgdH* (see below)”. Although the experimental data in this section confirm the associated RNA motifs to be guanidine riboswitches, the operon structure, comprising the genes for guanidine degradation, is also discussed.

Figure 1: Panels A and B should be moved to the Supplementary Information, as SI Figures 1 and 2 already display the same gels. Additionally, the two-dimensional structures of both riboswitches in the presence and absence of their respective ligands should be included in the SI for clarity.

We are convinced that the gels are necessary in order to illustrate the results and would like to keep them in the main part. As Nature Communications requires to provide full images of gels etc., we have added the full radiographs to the SI. We included two-dimensional structures to the SI to display the particular riboswitches of *V. boliviensis*. In the figure caption we also shortly address guanidine binding

by the respective riboswitch that has been studied in great detail before by in-line probing (Nelson et al., 2017; Sherlock et al., 2017), x-ray structure determination (Reiss et al., 2017; Battaglia et al., 2017; Huang et al., 2017, NMR studies (Schamber et al., 2022) and molecular dynamics simulations (Steuer et al., 2021 and 2024).

I am also somewhat puzzled by the kinetics shown, particularly in panel C. The normalized intensities appear unusually clean, even though the bands marked with red arrows in panel A seem to have similar intensities across all conditions. Could the authors clarify how these kinetics were derived and provide the data in the SI?

We agree that the changes in intensity are poorly visible by eye. The change is best observed by comparing the control lane (-) to the adjacent lane, where the sample with the highest guanidine concentration was loaded. The intensities are normalized to the lowest and highest intensities in order to highlight ligand sensitivity, a practice common when analysing in-line probing gels. We added a detailed description of how the data were obtained to the methods section: “Band intensities of modulated and unmodulated bands were determined using Quantity One (Bio-Rad). Modulated band intensity was referenced against the unmodulated band intensity in the respective lane. Intensities were normalized to the lowest and highest values and plotted against the ligand concentration. Data were fitted with non-linear regression assuming one binding site for Gd-I (Nelson et al.; Reiss et al.) and two binding sites for Gd-II (Sherlock et al.; Huang et al.). Thus, for Gd-II the Hill-equation was used for regression.” The raw data obtained from the radiograph are provided in agreement with Nature Communications policy in the source data file.

For panels C and D, do the graphs represent mean values from at least three independent replicates since KD values are given with \pm values? If so, the plots should include standard deviations, and the R^2 values for the fitted curves.

The data points represent single values from one experiment, a repetition resulted in similar results. The given errors are derived from the fit of the data points. We have clarified this issue by adding to the figure legend the phrase: Data points represent single measurements. Given errors in C and D represent standard errors derived from the fit, see methods for details. $R^2= 0.957$ (C); $R^2= 0,995$ and $R^2=0,996$ for guanidine and methylguanidine, respectively (D). Consistent results were obtained in independent experiments.

Line 132: The authors should consider renaming this subsection, as from my point of view MgdD is an oxygenase and not a demethylase.

We changed the name of MgdD to methylguanidine hydroxylase (MgdH) throughout the manuscript to better reflect the reaction catalysed by the enzyme. Accordingly, the header of the subsection was changed to “MgdH is a methylguanidine hydroxylase”

Lines 139-140: Can the authors comment on the reported K_m values of 150 μM for methylguanidine, 500 μM for 2-OG, and 240 μM for dimethylguanidine and clarify whether these concentrations are environmentally relevant? If not, the authors should address this discrepancy and discuss its implications in the Discussion section.

The value for 2-OG lies within the range of intracellular 2-OG concentrations reported in *E. coli* (doi: [10.1128/MMBR.00038-15](https://doi.org/10.1128/MMBR.00038-15)). As little is known about methylguanidine in nature, it is difficult to discuss the K_M values with respect to environmental relevance. Our work highlights the need for further research into guanidine and methylated guanidines in nature. However, even if the environmental concentrations should be below the K_M of our enzymes, it would not necessarily mean that it is not physiologically relevant. It would only mean that the enzyme is not completely saturated and turn-

over rates are slower. Thus, if the reaction is still fast enough to provide sufficient amounts of nitrogen to support the growth of *V. boliviensis* it should be considered relevant. In the discussion section, we also provide a brief statement on the K_M values and a short discussion on whether the ABC transporter encoded on the operon could act as a promiscuous guanidine/methylguanidine importer for accumulation of guanidine(s) as nitrogen sources.

Figure 3: The caption should include all abbreviations, such as NMMA and ADMA. The caption is somewhat misleading: it states that “data points represent single measurements” (in lines 163-164), yet K_m and k_{at} values are reported with \pm values. How can this be? In general, enzyme kinetics should be presented as mean values of replicates with corresponding standard deviations.

We now explain all abbreviations. The given standard errors derive from the regression curve. An independent replication of each experiment can be found in the source data file. We now included R^2 and mention that the fit was done using the Michaelis-Menten equation.

Additionally, the y-axis should show appropriate activity units, such as nkat/mg or U/mg, now it is in $\mu\text{mol/s/mg}$ (equivalent to $\mu\text{kat/mg}$).

We changed the title of the y-axis to $\mu\text{kat/mg}$.

Lines 184-185: The authors should specify the degree of conservation and clearly define the substrate binding motif, for example, in the format ...D49xR51....

We performed a more comprehensive bioinformatic analysis of MgdL addressing the conservation of the substrate binding pocket using the ConSurf server (Ashkenazy et al., 2016). Results are now included in the main text and illustrated in SI Figure 4D (see below).

Lines 182-191: The catalytic hypotheses presented are based on the predicted structure of MgdL (confidence score of the prediction should be added). However, additional experimental evidence is needed to support these claims. For instance, solving the crystal structure of MgdL in complex with the substrate and performing site-directed mutagenesis, particularly targeting the arginine residue proposed to act as an acid, would help validate the mechanism. As the authors themselves note, such a role for arginine is highly unusual and, to date, has only been observed in a serine recombinase, more data are needed here.

In response to this issue we performed the requested mutagenesis of proposed key residues, Asp49 and Arg51, involved in the proposed enzyme mechanism for MgdL and included it in the manuscript.

Mutation of both residues disrupted the catalytic activity of MgdL (SI Figure 5B). We added the MgdL structure prediction coloured by the local confidence score as SI Figure 2B (Chai et al., 2025). As the substrate N-hydroxymethylguanidine is unstable, it appears not feasible to get a structure of substrate-bound MgdL.

Lines 227-228: Do the authors have an explanation for why *V. boliviensis* failed to grow on dimethylguanidine, despite measurable in vitro activity with this substrate?

We included a short discussion into the results section of the growth experiments: “A potential reason for the observed growth deficiency is that dimethylguanidine is not a ligand for the Gd-II riboswitch and thus cannot enhance expression of *mgdH* and *mgdL* (SI Figure 2). Additionally, MgdL seemed not to accelerate the fragmentation of N-(hydroxymethyl)-N-methyl-guanidine and N,N-di(hydroxymethyl)-guanidine (SI Figure 6). Furthermore, dimethylguanidine might not be efficiently taken up by *V. boliviensis* if the substrate binding protein of the ABC transporter is selective for guanidine and methylguanidine.”

Could the release of a double amount of formaldehyde from dimethylguanidine be toxic to the strain?

We addressed the general issue of formaldehyde toxicity by re-analysing our proteome data and measuring formaldehyde in actively growing cultures of *V. boliviensis* on methylguanidine (see below). As there are other potential reasons why *V. boliviensis* failed to grow on dimethylguanidine besides formaldehyde toxicity (e.g. reduced uptake, poorer substrate for MgdL), we refrain from speculating about the reasons. We would like to point out the results with reduced (di)methylguanidine concentrations in the growth medium based on your suggestion (see next point) that confirmed our previous findings.

Additionally, did the authors attempt to cultivate the strain with lower concentrations of (dimethyl)guanidine, as 5 mM appears to be relatively high?

We repeated the experiment with methylguanidine and dimethylguanidine concentrations ranging from 0.2 to 2 mM. We observed residual growth of *V. boliviensis* for dimethylguanidine at 0.25 mM after 35 h. We included a short statement in the main text: “only residual growth could be observed after 35 h.” We include the data for 0.25 mM dimethylguanidine in Supplementary Figure 7, together with the negative control without nitrogen source.

Lines 225-236: This section is missing some important information, which is either addressed elsewhere in the manuscript (and should be referenced or moved here for clarity) or not addressed at all:

1) What happens to the formaldehyde produced in vivo? As a potent crosslinker, formaldehyde can be toxic to cells. Did the authors measure formaldehyde concentrations during growth? Was any accumulation observed? This aspect should be discussed in the text and, if available, integrated into the growth curve data.

The release of formaldehyde could indeed be toxic when large amounts of substrate are consumed for nitrogen assimilation. We checked our proteome data and found S-(hydroxymethyl)glutathione dehydrogenase and S-formylglutathione hydrolase involved in the detoxification of formaldehyde upregulated when *V. boliviensis* grew on methylguanidine. We included a statement regarding the potential toxicity and the putative detox system in our results part: “The formaldehyde released during the degradation of methylguanidine could be toxic to the cells. We did not detect formaldehyde in *V. boliviensis* cultures growing actively with methylguanidine as sole nitrogen source (data not shown, detection limit ≤ 10 μ M, SI Figure 8). Instead, we found that a probable formaldehyde detoxification system comprising predicted S-(hydroxymethyl)glutathione dehydrogenase (NCBI Acc.No.: KUC_2684)

and S-formylglutathione hydrolase (NCBI Acc.No.: KUC_2683) (Gonzalez et al.; 2006) was upregulated when *V. boliviensis* grew on methylguanidine (SI File 2)."

We tried to measure formaldehyde in cultures of *V. boliviensis* growing actively on methylguanidine. We were not able to detect formaldehyde in those cultures. When samples were spiked with 25 μ M formaldehyde before the Nash assay, formaldehyde could be detected. The induction of a formaldehyde detoxifying system in *V. boliviensis* could explain this observation.

2) Did the authors perform growth experiments using labeled nitrogen sources to track the incorporation of nitrogen from (methyl)guanidine into cellular biomass?

In our opinion, such an experiment would not add relevant new insights into the function of MgdH and MgdL. We show that *V. boliviensis* does not grow with only atmospheric N₂. The combination of MgdH and MgdL with enzymes that are known to convert guanidine into ammonium leaves little doubt that this ammonium will then be incorporated into all sorts of metabolites that are required to enable growth of the cells. We now show the negative control of the growth experiment that clearly illustrates that *V. boliviensis* cannot grow without additional nitrogen sources and hence is not able to use atmospheric N₂. We now mention the information in the main text: "The presence of only atmospheric N₂ did not stimulate any growth of *V. boliviensis* (SI Figure 7A)"

3) Why are MgdD and MgdL upregulated during growth on guanidine, even though guanidine is the product of their enzymatic activity? The explanation provided in lines 102–108 is helpful, but this rationale should also be included here for better coherence.

As guanidine is bound by the Gd-I and the Gd-II riboswitch, it seems to be the major regulator of the whole operon that is also comprising *mgdH* and *mgdL*. We included a short statement into the discussion: "The control by two types of guanidine responsive riboswitches with different substrate specificities and acting on transcription and translation could be an interesting subject for further studies."

Figure 4: In panel A, the growth curves should also include the concentrations of guanidine, methylguanidine, dimethylguanidine, or ammonium to demonstrate that growth was coupled with N-source consumption. Additionally, did the authors include a negative control without a N-source? If so, this data should be presented somewhere as well.

We repeated the growth experiment and subjected samples of *V. boliviensis* grown on 1 mM methylguanidine or guanidine to LC-MS analysis. As expected, both N-sources were consumed during growth of *V. boliviensis* (SI Figure 7). We included a negative control without a nitrogen source, as well as data for 0.25 mM dimethylguanidine as sole nitrogen source. Even at low concentrations, *V. boliviensis* was unable to efficiently utilise dimethylguanidine as nitrogen source (see above). Importantly, no growth was observed in absence of a nitrogen source.

It would also be valuable to include formaldehyde concentration measurements to assess potential accumulation during growth. These additional datasets could be shown in separate panels (e.g., panels B and C).

We addressed this question as described above. No formaldehyde was detectable in cultures of *V. boliviensis* growing on methylguanidine.

In the current panel B, the authors should clearly label what the rows represent, presumably log₂-transformed LFQ values?

We added the description of the data to the panel.

Discussion:

The reported KD values, particularly for the second riboswitch, at approximately 90 μM , are relatively high. The same applies to the KM values for methylguanidine ($\sim 150 \mu\text{M}$) and dimethylguanidine ($\sim 240 \mu\text{M}$). If I am correct, environmental concentrations of (di)methylguanidine are typically in the low micromolar (single-digit μM) range. Do the authors have access to alternative environmental concentration data, particularly for habitats where these halophilic strains are found? Could local concentrations be higher in such niches?

Alternatively, might there be a dedicated transporter chaperone that facilitates the uptake of methylguanidine, thereby increasing local intracellular concentrations and effectively lowering the apparent KD or KM (meaning that MgdD does not interact with the free methylguanidine)?

The authors should address this discrepancy in detail and consider including a discussion of possible transport mechanisms or environmental conditions that could reconcile the observed affinities with ecological relevance.

Unfortunately, we do not have access to any additional environmental data. However, it seems plausible that methylguanidine accumulates in extreme habitats such as the salt lake from which *V. boliviensis* was isolated. This is especially likely given that methylguanidine has been shown to be produced abiotically by radical oxygen species, as discussed.

We can only speculate about the substrate specificity of the ABC-transport system that is located in the (methyl)guanidine utilisation operon. The presence of a predicted periplasmic binding protein and an ATP-dependent import system implies that intracellular concentrations of (methyl)guanidine might be much higher than in the environment. As mentioned above we now address the points raised in the discussion section. In addition, we added further information concerning the substrate specificity of guanidine transporters. Although importers have not been studied yet, it has been shown that guanidine exporters also accept methylguanidine as substrate (Kermani et al., 2028).

Minor review

Main text:

Line 57: “followed by” – typo- Changed

Line 68: “gca gene” – genes must be italic. Changed

Line 106: SI Figure 2 is cited, but SI Fig. 1 was not cited yet.

Line 134: Which *E. coli* strain was used? BL21 (DE3) was added.

Lines 168 and 170: “m/z” – must be italic. Changed

Line 191: A new paragraph should begin after reference 26 to improve readability and structure. Changed

Figure 3: Explain all abbreviations used in the figure, such as 2OG, SA. We added the sentence “The co-substrate 2-oxoglutarate (2OG) is converted to succinate (SA)” to the figure legend.

Supplementary information:

SI Figure 4: For improved clarity, the substrate should be highlighted in a different colour. We changed the colour of the protein backbone to improve the clarity, as we would like to keep the substrate colour that matches the colour scheme throughout the manuscript.

Line 42: “form” – typo. **Changed**

Line 50: “LC-MS” – typo. Explain the DMGd abbreviation. **Changed and substituted by dimethylguanidine.**

SI Figure 7: The manuscript does not contain a citation or reference to this figure in the main text.

SI Figure 7 is referenced in the results section about the Nash reaction for the detection of formaldehyde.

SI Table 1: For improved clarity, the authors should clearly indicate which substrates were well transformed, poorly transformed, or not transformed at all using colour coding, symbols (e.g., +/-), or numerical values representing specific activities.

Substrates of MgdH are now highlighted in green.

References:

It would be advisable for the authors to review the formatting of the references. I noticed that some references have titles written in uppercase letters, while others do not (e.g., lines 383-389 vs. 380-381 but there are much more), and also abbreviated journal names are formatted differently.

We corrected the formatting of the references.

Reviewer 2:

This manuscript by Sinn et al. and Hartig describes the identification and validation of two novel enzymes for the oxidation of methylguanidine and the decomposition of the resulting product into guanidine and formaldehyde. The enzymes were identified by their association with two classes of guanidine-binding riboswitches, and this genomic analysis approach is proving to be a powerful way to link unusual proteins with their biochemical functions via riboswitch association. Their findings also expand the list of enzymes that manipulate guanidine or its close chemical derivatives, and help reveal previously unknown ways in which guanidine is involved in the metabolism and physiology of many species. The importance of this and related work is likely to increase as more discoveries are made. The work appears to be carefully done, the results appear to be sound, and the manuscript is generally well written. I only have very minor comments that the authors might wish to consider when preparing their manuscript for publication. Importantly, none of these comments should be used to hold up publication.

Minor Comments

1. The title is a bit obscure and will be uninteresting to most readers. I recommend putting ‘guanidine’ in the title. **We changed the title to be more precise: “Demethylation of methylguanidine by a stepwise dioxygenase and lyase reaction”**

2. Line 37: Write “and” in place of “end”. **Changed**

3. Line 99: Flip references 23 and 24 so that the in-line probing method can be cited on its first mention in the text. **Changed**

4. Fig. 1: Make gene names white when placed in a dark-colored arrow. **Changed**

5. Line 101: Delete the extra space after “below”. **Changed**

6. Fig. 2A: Band changes are not clear on visual inspection, such that it is not clear if the arrows are in the right locations. The same is true for SI Figure 1. It would be good if the authors commented on the lack of obvious changes in band intensity.

We agree that the changes in intensity is poorly visible by eye. The change is best observed by comparing the control lane (-) to the adjacent lane, where the sample with the highest guanidine concentration was loaded. We included a short statement in the figure caption of Figure 2, see also response to reviewer 1.

7. Also in the gel figures, the authors use -OH and T1 nuclease to generate marker bands, but they don't use these to label the band locations to aid the reader. I don't think this causes doubt about the interpretations/conclusions, but it would be great if bands could be annotated.

We now included labels and show annotated secondary structures of the Gd-I and Gd-II riboswitches in SI Figure 1A and SI Figure 2A, respectively.

8. Legend to Fig. 4, second to last line: Remove the dash from "Hill-equation". Changed

9. Line 151: Remove the dash from "Nash-reagent". Changed

10. Line 159: The "D/E/F" notation is different from other legends.

11. Line 182: Delete the space between "Chai-1" and "25". Changed

12. Line 226 and elsewhere: Write "as a sole nitrogen source" in place of "as sole nitrogen source". Changed

13. Line 278: Delete the extra space between "electrons" and "28". Changed

14. References: Use the journal format for reference titles (for example, see the difference between Reference 1 and 2 regarding the use of upper-case letters). Changed

15. SI Figure 5: I have a minor difference of opinion regarding the proposed reaction mechanism. If I were designing a fast enzyme, I would have the positively charged nitrogen of R51 pointing a proton at the nitrogen atom bonded to the hydroxymethyl group. The positively charged nitrogen would more easily donate a proton than the adjacent NH₂ group.

As the electrons and thus the charge of the guanidine moiety is presumably delocalized, the depicted resonance structure of the guanidinium group in the proposed mechanism would equal the suggested situation. Since delocalization might be hampered in the active site and due to didactic considerations, we have followed the suggestion and placed the charged NH₂ adjacent to the bridging N in methylguanidine (Figure 4).

Reviewer 3:

The manuscript describes the characterization of two new enzymes, MgdD and MgdL, whose genes were found in the guanidine degradation operon in *V. boliviensis*. MgdD was shown to be a 2-oxoglutarate/Fe(ii)-dependent dioxygenase that hydroxylates methylguanidine. The stable N-hydroxymethylguanidine is converted to guanidine and formaldehyde by the newly-discovered N-hydroxymethylguanidine lyase MgdL. The study is interesting and reveals the mechanism for an important biochemical reaction. The manuscript is well written, and the study is well developed. I did not notice any major flaws in methodology or data analysis. However, there are a few items that the authors should address:

1. Lines 67-70: The authors mention that the guanidine operon in *V. boliviensis* is similar to guanidine operons in other bacteria, but with two additional genes (mgdD and mgdL). How similar are the other

genes in the operon to operons from other species? It would be helpful to perform a more in-depth bioinformatic analysis between the *V. boliviensis* operon and homologous operons from other species. Further, are the genes all in the same orientation? Organized in the same order? Are all of the other operons the same? The authors could consider including other operons in Figure 1 to compare with the operon in this study.

We searched for homologous operons by cblaster (Gilchrist et al., 2021) using the CAGECAT server (<https://cagecat.bioinformatics.nl/>). We found guanidine carboxylase operons including MgdH and MgdL homologs only in *Vreelandella* and *Salinisphaera* species, whereas *gca* operons lacking such genes are broadly distributed across many bacterial taxa (Schneider et al.; 2020). The extended operon structure in other *Vreelandella* and *Salinisphaera* is very similar. In *Salinisphaera*, *mgdH* and *mgdL* reside downstream of *atzF*. However, the orientation of all genes is the same. *Salinisphaera hydrothermalis* has a sequence identity of 70%, 75%, 71%, 69%, 46% and 64% for *cgdA*, *cgdB*, *gca*, *mgdH*, *mgdL* and *atzF* of *V. boliviensis*, respectively. Although we agree that an in-depth analysis of the probable evolution of the *gca* operons is an interesting topic, we feel that a more extensive comparison of operon structures and sequence similarities would distract from the main focus of the present manuscript, which is the biochemical characterisation of the MdgH/MgdL enzyme system.

2. Figure 1: the *atzF* black text in the purple arrow is difficult to read. **Changed**

3. Lines 106-108: The authors should demonstrate that the Gd-I riboswitch acts through transcriptional control and the Gd-II riboswitch acts through translational control.

Both types of riboswitches have been studied in great detail and evidence for transcriptional or translational control have been presented in the past, e.g. in the references that are cited within the manuscript. Additional evidence for the riboswitch expression platform mechanism in the described operon seems out of the scope of the manuscript. We now added a short sentence and additional citations about the mechanism of Gd-II (“However, it has been shown that regulation of mRNA levels and translational control can be tightly linked in riboswitch-regulated genes.^{25,26}”) and included a short statement that the unusual control by two different riboswitches might be an interesting subject for further studies: “The control by two types of guanidine responsive riboswitches with different substrate specificities acting on transcription and translation could be an interesting subject for further studies.”

4. Line 152: Should the callout for Figure 3C be Figure 2C? **Changed**

5. Figure 2 legend: Please define DMGd and describe the reaction in the legend. I see a description for the graphs in parts D and E, but no information for part F. **Included**

6. SI Figure 4 legend (line 42): “E126 and D112 could form hydrogen bonds...” Should the “for” be “form”? **Changed**

We would like to thank all reviewers again for the careful reading of our manuscript.

Reviewer 1:

I have re-reviewed the manuscript by Sinn et al., now entitled "Demethylation of methylguanidine by a stepwise dioxygenase and lyase reaction" (previously: "Demethylation catalyzed by a step-wise dioxygenase and lyase tandem reaction").

The authors have addressed all of my previous comments and incorporated the additional data I requested in a satisfactory manner. The manuscript has improved significantly as a result of the revision.

However, I did notice a few minor typographical and/or formatting errors that should be corrected prior to publication.

Line 90: For consistency, "MgdD" should be changed to MgdH. **Changed.**

Line 113: "Fig. 2D" must be changed to 3D. **Changed.**

Line 114: "Fig. 2E" must be changed to 3E. **Changed.**

Line 116: "Fig. 2F" must be changed to 3F. **Changed.**

Line 125: "Fig. 2C" must be changed to 3C. **Changed.**

Line 145: "m/z" – must be italic. **Changed**

Line 149: Which E. coli strain was used? We added the specific strain in the text "BL21 (DE3)".

Lines 165 and 172: "D48" must be changed to D49. **Changed.**

Reviewer 2:

The authors have addressed all the original minor concerns I noted, and all looks good to proceed with publication.

Reviewer 3:

The authors have adequately addressed my concerns.

In this study, Sinn and colleagues uncover a novel guanidine metabolism pathway in *Vreelandella boliviensis*, regulated by guanidine-responsive riboswitches. Researchers identified two previously uncharacterized proteins within a riboswitch-controlled operon: (i) a 2-oxoglutarate/Fe(II)-dependent dioxygenase MgdD, which hydroxylates methylguanidine to N-hydroxymethylguanidine; and (ii) an N-(hydroxymethyl)guanidine lyase MgdL that accelerates its breakdown into guanidine and formaldehyde. The authors describe that *V. boliviensis* can use either guanidine or methylguanidine as its sole nitrogen source, claiming that these findings reveal two new enzymatic activities and expand the understanding of guanidine assimilation and regulation in halophilic bacteria.

Although the results of this study, mainly resulting from activity assays with recombinant MgdD and MgdL, structural predictions, mass spectrometry and growth experiments using either ammonia or guanidine/methylguanidine/dimethylguanidine look interesting, additional in-depth analyses are needed to enhance the impact and quality of the publication. Moreover, there are several aspects that require clarification. These points will be addressed in the following sections.

Major review

Title:

The authors should consider revising the title, as it is currently too broad and lacks scientific precision.

Abstract:

Line 22-23: This statement is not entirely accurate, as Hausinger et al. already reported the broad activity of 2-OG/Fe(II) dioxygenases in 2015 (doi: 10.1039/9781782621959-00001). The authors should consider softening this claim to reflect prior findings.

Results:

Line 82: The authors should consider renaming this subsection, as it primarily focuses on the characterization of the two riboswitches and their kinetics.

Figure 1: Panels A and B should be moved to the Supplementary Information, as SI Figures 1 and 2 already display the same gels. Additionally, the two-dimensional structures of both riboswitches in the presence and absence of their respective ligands should be included in the SI for clarity. I am also somewhat puzzled by the kinetics shown, particularly in panel C. The normalized intensities appear unusually clean, even though the bands marked with red arrows in panel A seem to have similar intensities across all conditions. Could the authors clarify how these kinetics were derived and provide the data in the SI?

For panels C and D, do the graphs represent mean values from at least three independent replicates since K_D values are given with \pm values? If so, the plots should include standard deviations, and the R^2 values for the fitted curves.

Line 132: The authors should consider renaming this subsection, as from my point of view MgdD is an oxygenase and not a demethylase.

Lines 139-140: Can the authors comment on the reported K_m values of 150 μM for methylguanidine, 500 μM for 2-OG, and 240 μM for dimethylguanidine and clarify whether these concentrations are environmentally relevant? If not, the authors should address this discrepancy and discuss its implications in the Discussion section.

Figure 2: The caption should include all abbreviations, such as NMMA and ADMA. The caption is somewhat misleading: it states that “data points represent single measurements” (in lines 163-164), yet K_m and k_{at} values are reported with \pm values. How can this be? In general, enzyme kinetics should be presented as mean values of replicates with corresponding standard deviations. Additionally, the y-axis should show appropriate activity units, such as nkat/mg or U/mg, now it is in $\mu\text{mol/s/mg}$ (equivalent to $\mu\text{kat/mg}$).

Lines 184-185: The authors should specify the degree of conservation and clearly define the substrate binding motif, for example, in the format ...D₄₉XR₅₁....

Lines 182-191: The catalytic hypotheses presented are based on the predicted structure of MgdL (confidence score of the prediction should be added). However, additional experimental evidence is needed to support these claims. For instance, solving the crystal structure of MgdL in complex with the substrate and performing site-directed mutagenesis, particularly targeting the arginine residue proposed to act as an acid, would help validate the mechanism. As the authors themselves note, such a role for arginine is highly unusual and, to date, has only been observed in a serine recombinase, more data are needed here.

Lines 227-228: Do the authors have an explanation for why *V. boliviensis* failed to grow on dimethylguanidine, despite measurable *in vitro* activity with this substrate? Could the release of a double amount of formaldehyde from dimethylguanidine be toxic to the strain? Additionally, did the authors attempt to cultivate the strain with lower concentrations of (dimethyl)guanidine, as 5 mM appears to be relatively high?

Lines 225-236: This section is missing some important information, which is either addressed elsewhere in the manuscript (and should be referenced or moved here for clarity) or not addressed at all:

- 1) What happens to the formaldehyde produced *in vivo*? As a potent crosslinker, formaldehyde can be toxic to cells. Did the authors measure formaldehyde

concentrations during growth? Was any accumulation observed? This aspect should be discussed in the text and, if available, integrated into the growth curve data.

2) Did the authors perform growth experiments using labeled nitrogen sources to track the incorporation of nitrogen from (methyl)guanidine into cellular biomass?

3) Why are MgdD and MgdL upregulated during growth on guanidine, even though guanidine is the product of their enzymatic activity? The explanation provided in lines 102–108 is helpful, but this rationale should also be included here for better coherence.

Figure 4: In panel A, the growth curves should also include the concentrations of guanidine, methylguanidine, dimethylguanidine, or ammonium to demonstrate that growth was coupled with N-source consumption. Additionally, did the authors include a negative control without a N-source? If so, this data should be presented somewhere as well. It would also be valuable to include formaldehyde concentration measurements to assess potential accumulation during growth. These additional datasets could be shown in separate panels (e.g., panels B and C).

In the current panel B, the authors should clearly label what the rows represent, presumably log₂-transformed LFQ values?

Discussion:

The reported K_D values, particularly for the second riboswitch, at approximately 90 μM , are relatively high. The same applies to the K_M values for methylguanidine (~150 μM) and dimethylguanidine (~240 μM). If I am correct, environmental concentrations of (di)methylguanidine are typically in the low micromolar (single-digit μM) range. Do the authors have access to alternative environmental concentration data, particularly for habitats where these halophilic strains are found? Could local concentrations be higher in such niches?

Alternatively, might there be a dedicated transporter chaperone that facilitates the uptake of methylguanidine, thereby increasing local intracellular concentrations and effectively lowering the apparent K_D or K_M (meaning that MgdD does not interact with the free methylguanidine)?

The authors should address this discrepancy in detail and consider including a discussion of possible transport mechanisms or environmental conditions that could reconcile the observed affinities with ecological relevance.

Minor review

Main text:

Line 57: “followed by” – typo.

Line 68: “*gca* gene” – genes must be italic.

Line 106: SI Figure 2 is cited, but SI Fig. 1 was not cited yet.

Line 134: Which *E. coli* strain was used?

Lines 168 and 170: “*m/z*” – must be italic.

Line 191: A new paragraph should begin after reference 26 to improve readability and structure.

Figure 3: Explain all abbreviations used in the figure, such as 2OG, SA.

Supplementary information:

SI Figure 4: For improved clarity, the substrate should be highlighted in a different colour.

Line 42: “form” – typo.

Line 50: “LC-MS” – typo. Explain the DMGd abbreviation.

SI Figure 7: The manuscript does not contain a citation or reference to this figure in the main text.

SI Table 1: For improved clarity, the authors should clearly indicate which substrates were well transformed, poorly transformed, or not transformed at all using colour coding, symbols (e.g., +/-), or numerical values representing specific activities.

References:

It would be advisable for the authors to review the formatting of the references. I noticed that some references have titles written in uppercase letters, while others do not (e.g., lines 383-389 vs. 380-381 but there are much more), and also abbreviated journal names are formatted differently.